*Report*

# DeepCycle reconstructs a cyclic cell cycle trajectory from unsegmented cell images using convolutional neural networks

Luca Rappez[1,2], Alexander Rakhlin[3], Angelos Rigopoulos[1], Sergey Nikolenko[4,5] & Theodore Alexandrov[1,6,*]

## Abstract

The advent of single-cell methods is paving the way for an in-depth understanding of the cell cycle with unprecedented detail. Due to its ramifications in nearly all biological processes, the evaluation of cell cycle progression is critical for an exhaustive cellular characterization. Here, we present DeepCycle, a deep learning method for estimating a cell cycle trajectory from unsegmented single-cell microscopy images, relying exclusively on the brightfield and nuclei-specific fluorescent signals. Deep-Cycle was evaluated on 2.6 million single-cell microscopy images of MDCKII cells with the fluorescent FUCCI2 system. DeepCycle provided a latent representation of cell images revealing a continuous and closed trajectory of the cell cycle. Further, we validated the DeepCycle trajectories by showing its nearly perfect correlation with real time measured from live-cell imaging of cells undergoing an entire cell cycle. This is the first model able to resolve the closed cell cycle trajectory, including cell division, solely based on unsegmented microscopy data from adherent cell cultures.

**Keywords** cell cycle; deep learning; live-cell imaging; single-cell analysis; trajectory inference

**Subject Categories** Cell Cycle; Computational Biology

**Mol Syst Biol. (2020) 16: e9474**

## Introduction

Single-cell technologies, including microscopy, have emerged as methods of choice to reveal cellular biology at increasing temporal and spatial resolutions. A key challenge of our time is to interpret a large amount of single-cell data and put it into the context of current biological knowledge. Brightfield and fluorescence microscopy have been exploited for decades to gather spatial multiplexed information about cells. Using genetically encoded systems for fluorescent reporter proteins, such as those used in the FUCCI2 system for characterization of the cell cycle, has transformed the field of cell biology by providing a real-time single-cell microscopy-compatible readout (Sakaue-Sawano *et al*, 2008). However, a key limitation of such approaches is that they require the cells to be transformed with a targeted fluorescent system, which only reveals the abundance or activity of a few known proteins.

In parallel, there is a growing interest in filling our gaps of knowledge about the cell cycle, the phenomenon integrating oscillating biological processes from the genesis of a cell until its division. This interest stems from recently discovered links of cell cycle with other cell biology phenomena such as metabolism (Cai & Tu, 2012) as well as its importance in disease such as cancer (Malumbres & Barbacid, 2009; Szczerba *et al*, 2019), but also from the recognition of cell cycle being a critical confounding factor in single-cell analyses (McDavid *et al*, 2016). With the increased use of spatially resolved single-cell analyses, particularly microscopy (Bove *et al*, 2017), this has motivated recent work on inferring the progression of a cell along its cell cycle solely from microscopy images, without explicitly labeling the cell cycle markers. Gut *et al* (2015) proposed a method for approximating cell cycle phases from fixed adherent cultures based on features obtained from cell segmentation. This approach, however, puts strong requirements on the cells (fixation, high-resolution microscopy, use of several fluorescent channels) that make them hard to combine with other types of single-cell analyses, in particular to investigate dynamics of the cell cycle via, e.g., live or time-lapse microscopy. Eulenberg *et al* (2017) demonstrated the power of deep learning to automatically extract features from microscopy images by reconstructing a cell cycle trajectory from flow-through microscopy images of individual cells. While flow-through microscopy is a widespread and

---

1  Structural and Computational Biology Unit, European Molecular Biology Laboratory, Heidelberg, Germany
2  Faculty of Biosciences, Collaboration for joint PhD degree between EMBL and Heidelberg University, Heidelberg, Germany
3  Neuromation OU, Tallinn, Estonia
4  National Research Institute Higher School of Economics, St. Petersburg, Russia
5  Steklov Institute of Mathematics at St. Petersburg, St. Petersburg, Russia
6  Skaggs School of Pharmacy and Pharmaceutical Sciences, University of California San Diego, La Jolla, CA, USA
   *Corresponding author. Tel: +49 06221 387 8690; E-mail: theodore.alexandrov@embl.de

accepted technology, it cannot investigate the spatial organization of cells and cell–cell interactions, phenomena known to be involved in the cell cycle (Gut et al, 2015).

In light of these limitations, we have developed DeepCycle, an open source deep neural network able to reconstruct a continuous closed cell cycle trajectory solely based on unsegmented microscopy images of adherent cells that only contain the brightfield and Hoechst channels. To train DeepCycle, we used single MDCKII cell fluorescent readouts of mAG-Geminin (1–110) and mKO2-Cdt1 (30–120), fluorescent proteins encoded in the FUCCI2 system (Sakaue-Sawano et al, 2008; Streichan et al, 2014). We developed a novel embedding strategy where cells are grouped into several virtual classes according to their relative intensities of mAG-Geminin and mKO2-Cdt1 markers used in FUCCI2. This provides discrete labels which were used as classification objectives during the network training. Ultimately, DeepCycle provides for each cell a low-dimensional vector which is used to visualize the relationships between the cells using dimensionality reduction methods. With this approach, we demonstrated the feasibility of reconstructing a cell cycle trajectory of adherent cells from unsegmented images. Finally, using live-cell imaging, we validated the DeepCycle representation by showing a nearly perfect correlation between DeepCycle pseudotime and the relative progression of real time through the cell cycle (CC time) measured from cells undergoing a whole cell cycle. We expect DeepCycle to enable the combination of label-free cell cycle investigation with pre-existing microscopy-related technologies involving conventional fluorescent dyes. We also anticipate DeepCycle to be used together with other cellular assays such as single-cell metabolomics where it can help identify and compensate for the confounding influence of cell cycle on the molecular readouts.

## Results and Discussion

In order to train the DeepCycle model, we imaged unsynchronized MDCKII cells with the FUCCI2 plasmid for over 33 h (200 acquisitions separated by 10 min) (Fig 1A; Streichan et al, 2014). This generated about 2.6 million cell images in four channels (brightfield, Hoechst, mAG, mKO2) that were split into two groups. First, the brightfield and Hoechst channels were used as input for the DeepCycle model. Second, the mAG and mKO2 channels were used

for the FUCCI2 mAG-Geminin and mKO2-Cdt1 markers, respectively, which provide a continuous proxy of the cells' progression through their cell cycle and were used to create the ground truth for the model training (Sakaue-Sawano et al, 2008). The Deep-Cycle model is a custom design of a deep convolutional neural network classifier (Fig 1B; for a more detailed description of the DeepCycle model, see Appendix Fig S1). A brightfield and Hoechst two-channel image of an unsegmented single cell is taken as input to the network and passed through five convolutional blocks where it is progressively downscaled. The convolutional part is followed by global average pooling and two fully connected layers with 512 and 4 neurons, respectively. For each image, softmax activation is applied after the last fully connected layer in order to output its probability to belong to the four virtual classes. The virtual classes are defined as pseudo-quadrants of the two-dimensional space of the mAG-Geminin and mKO2-Cdt1 fluorescence intensities which have been balanced for cell count (Fig 1C). Defining the virtual classes and using them in the objective function for our deep neural network is the key step that enabled the reconstruction of the cell cycle trajectory. The idea behind training with virtual classes is to tackle the lack of labeled cell cycle phases conventionally used for supervised training. Indeed, other works on cell cycle trajectory reconstruction rely on labeled cell phases (Eulenberg et al, 2017) or infer phase labels from data (Gut et al, 2015). Instead, we assign cells to virtual classes based on the FUCCI2 marker fluorescence which reflects the continuous cell cycle progression and, compared to cell cycle phases, does not rely on manual labeling. The accuracies for predicting the virtual classes are reported in Appendix Fig S2. Pilot experiments aiming to predict manually labeled cell cycle phases instead of unsupervised virtual classes are discussed in the DeepCycle framework design section of the Methods. The relevance of using a 2-channel input image as well as the reliance on unsegmented images is also discussed.

Next, we evaluated how the learned four-dimensional representation of each cell is related to cell cycle phases. Applying the Uniform Manifold Approximation and Projection (UMAP) algorithm to feature vectors of individual cells, we mapped 2.6 million cell images into a two-dimensional space (Fig 1D). UMAP is an unsupervised visualization approach representing the cells with similar DeepCycle features closer to each other. We show that UMAP represents all cells in a closed and almost cyclic structure.

**Figure 1. DeepCycle reconstructs a cell cycle closed trajectory from single-cell microscopy images.**

A  A tiled time-lapse microscopy imaging of MDCKII cells with the FUCCI2 system was performed over the area of approximately 5 × 5 mm for 33 h followed by automated cell tracking, generating 2.6 million individual cell images. For each track, the brightfield and Hoechst channels of unsegmented cell images centered at the nucleus were used as input for the DeepCycle deep learning network. Four virtual classes were defined by relative intensities of the FUCCI2 channels (mKO2-Cdt1 and mAG-Geminin) and used as ground truth for training the DeepCycle network. For contrast enhancement in this figure, the fluorescent intensities of each miniature single-cell image were clipped at their 70 percentile.

B  The architecture of the DeepCycle convolutional neural network. A 2-channel cell image is transformed into 3-channel and fed into ResNet-34 pre-trained on ImageNet. Then, a 256 feature map is generated by a conv4 block global average pooling followed by a fully connected layer and a softmax layer generating the probabilities of a cell to belong to each of the four virtual classes illustrated in (C). The values obtained before the softmax layer are used as features for a low-dimensional representation of the cells as visualized in (D).

C  Virtual classes of cells derived from the FUCCI2 intensities (mKO2 and mAG) which serve as ground truth for training the neural network (classification accuracies are reported in Appendix Fig S2). The virtual classes were designed to have similar numbers of cells and represent cells with different combinations of mAG and mKO2 intensities.

D  The DeepCycle low-dimensional representation of the cells. Coloring cells by their corresponding FUCCI2 expression profile (green: mAG$^+$/mKO2$^-$; red: mAG$^-$/mKO2$^+$; yellow: mAG$^+$/mKO2$^+$) reveals a continuous and cyclic progression through the cell cycle.

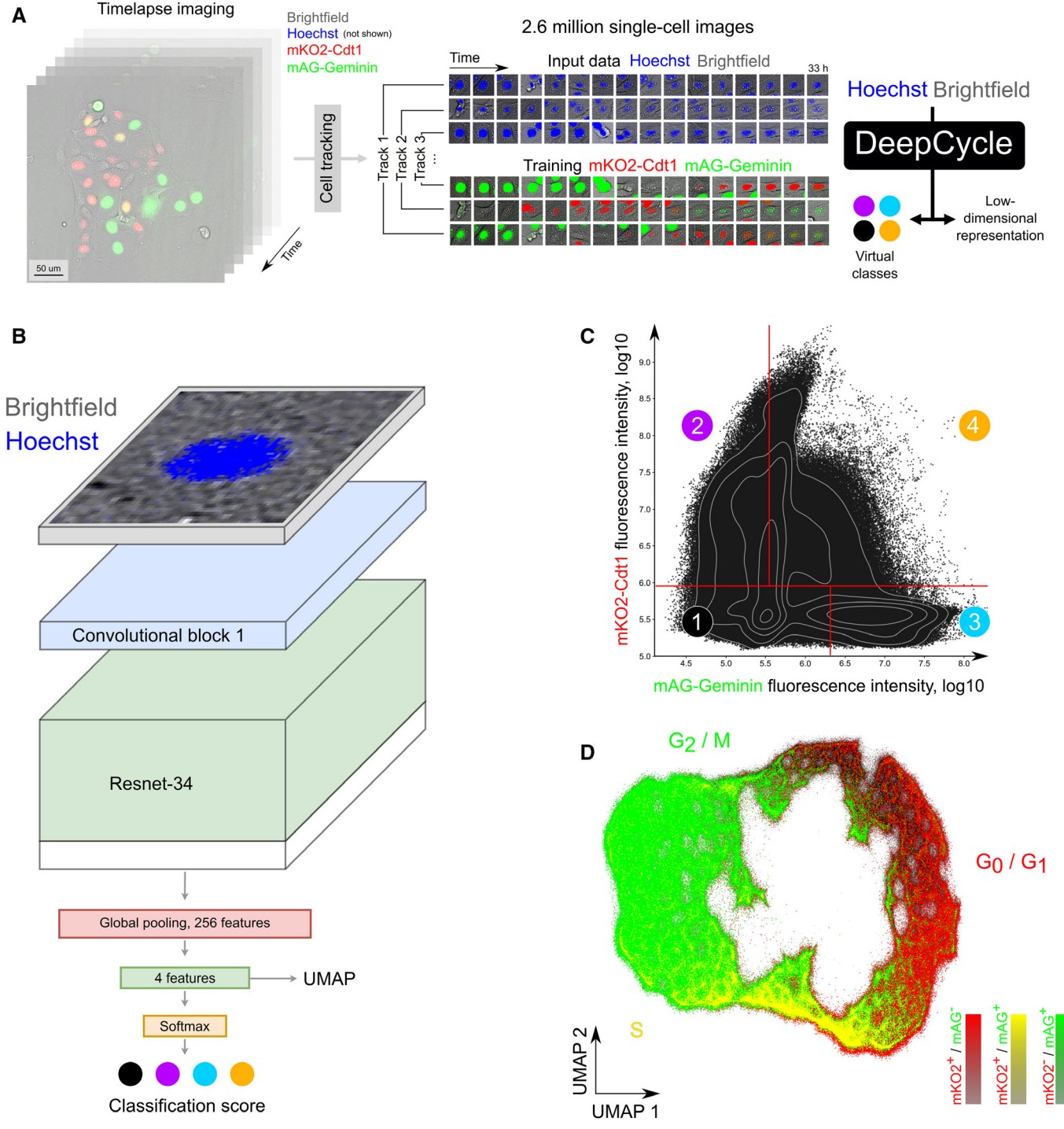

**Figure 1.**

In order to demonstrate that the learned representation indeed organizes cells along their cell cycle progression, we colored the cells with their respective FUCCI2 expression profile. Cells with the highest intensity for mKO2-Cdt1 (red), characteristic of the $G_0/G_1$ phases, group together in the upper right part. Cells with the double expression of mKO2-Cdt1 and mAG-Geminin (yellow) defining the early S phase are located at the bottom of the UMAP representation. Finally, cells with the highest mAG-Geminin intensity, characteristic of the late $S/G_2/M$ phases, group in the left part.

Earlier, similar efforts using flow-through microscopy have demonstrated the reconstruction of a continuous cell cycle trajectory (Eulenberg *et al*, 2017). However, cell cycle trajectories in both Eulenberg *et al* and Gut *et al* were linear and interrupted at the M phase (cell division point), whereas our method improves over these findings by capturing a continuous trajectory even through the

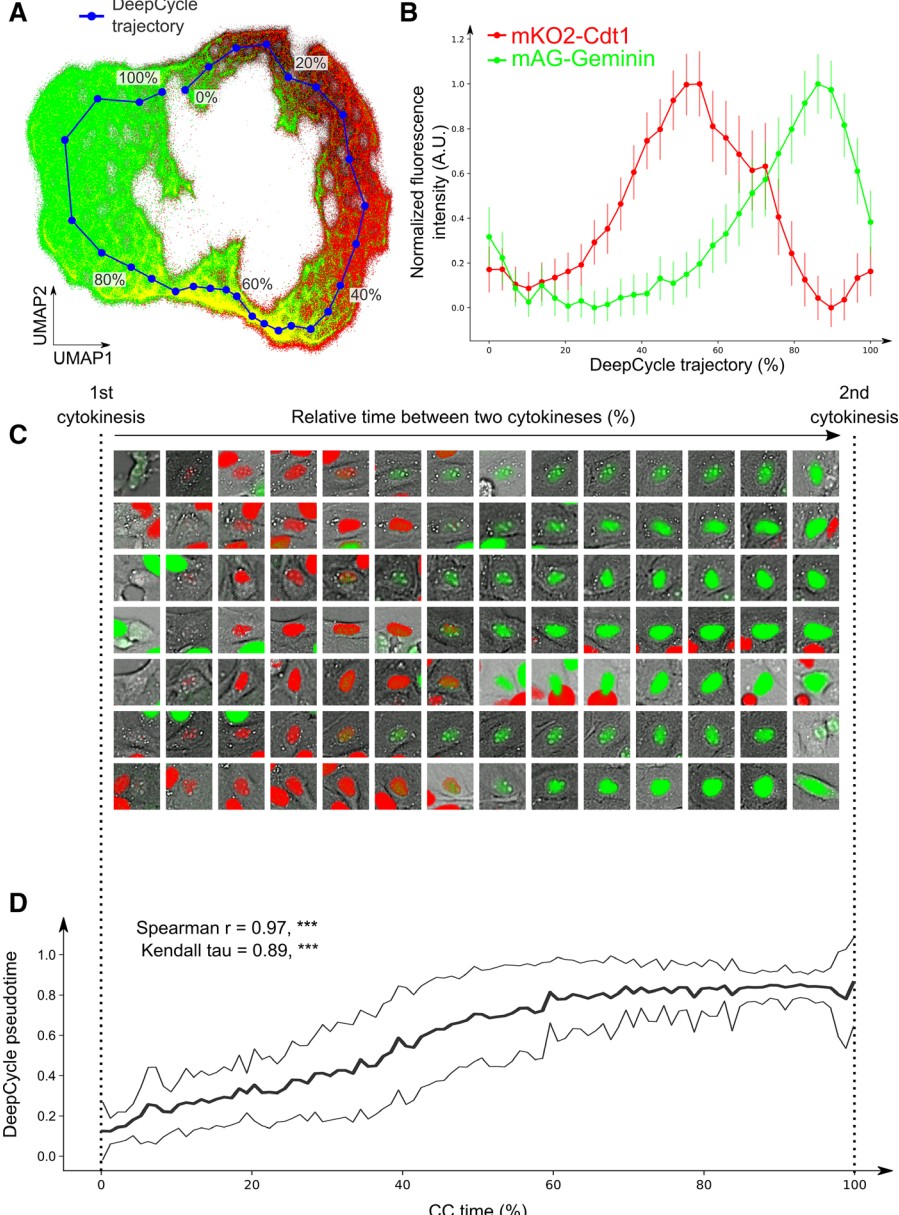

**Figure 2. DeepCycle recovers the relative real-time progression along the cell cycle.**

A   The DeepCycle trajectory (blue) goes through the reconstructed latent representations of cells.

B   Single-cell intensities of the FUCCI2 fluorescence markers along the DeepCycle trajectory. Scaled mean fluorescence intensities from mAG-Geminin (green) and mKO2-Cdt1 (red) are plotted; error bars show standard deviations (average $n$ = 89,481; total $n$ = 2,684,453).

C   In order to validate the DeepCycle pseudotime, we manually selected 50 cells that have been tracked over an entire cell cycle, as defined by the time spent between two successive cytokineses (CC time). As an illustration, representative cells are displayed with the overlay of their FUCCI2 fluorescence intensities (green: mAG-Geminin; red: mKO2-Cdt1). The images are arranged along their CC time, where the first image is taken just after the first cytokinesis, and the last image is taken right before the second one. In each miniature cell image, the fluorescence intensities have been clipped at the top 70[th] percentile for visual clarity.

D   The DeepCycle pseudotime highly correlates with the CC time for the 50 cells tracked through their cell cycle (bold and thin black lines represent the average and the standard deviation, respectively). These cells were not part of the training set for DeepCycle. The high values of the Spearman and Kendall tau correlation coefficients (0.97 and 0.89, respectively, two-sided $P$-values < 0.001, ***, $n$ = 50) validate the ability of DeepCycle to reconstruct the relative progression of cells along their cell cycle solely from microscopy images.

division. Reconstructing a closed trajectory demonstrates continuity between the mother cell and its daughter cells which is captured by the microscopy. Thus, our results illustrate the potential of advanced image representation methods, in particular deep learning, to learn continuous trajectories from large populations of unsynchronized adherent cells exhibiting the full spectrum of cell cycle phases.

To further investigate how FUCCI2 fluorescence intensities change along the closed trajectory, we computed a DeepCycle trajectory as a one-dimensional projection of the DeepCycle feature vectors from the 2.6 million cell images (Fig 2A). We then defined the DeepCycle pseudotime as the progression, from 0 to 100%, of its trajectory from beginning to end. Figure 2B shows the average fluorescence intensities of the mKO2-Cdt1 and mAG-Geminin markers from all single-cell images along the DeepCycle pseudotime which coincide with fluorescence trends of the FUCCI2 system (Sakaue-Sawano *et al*, 2008) (Appendix Fig S3A for our FUCCI2 measurements). This provides quantified evidence of the capacity of DeepCycle to organize the cells along their cell cycle progression solely based on single-cell microscopy images in the Brightfield and Hoechst channels.

The average FUCCI2 time trends obtained from a cell population aligned around their cytokinesis event are highly variable (Appendix Fig S3A). This is partly explained by the well-known heterogeneity of cellular division rates (Sandler *et al*, 2015) but also due to variations in their commitment to continuously divide (Spencer *et al*, 2013; Sandler *et al*, 2015; van Velthoven & Rando, 2019). Indeed, we have observed that the FUCCI2 intensities of a population of 1,024 dividing cells become more heterogeneous after cell division (Appendix Fig S3B). This could be explained by the de-commitment of some cells to re-enter the cell cycle, as suggested by the bimodality of both mKO2-Cdt1 and mAG-Geminin intensities where the cells committed to divide are characterized by higher cell cycle marker expression (Appendix Fig S3C). In addition, we have been able to estimate that the division rate of MDCKII cells ranged from 20 to 32 h in our experiment (min–max, over 50 cells). We believe that this also contributes to the heterogeneous FUCCI2 intensities observed across the whole population when projected on real-time progression. To compensate for variable division rates and the presence of quiescent cells, we manually selected 50 cells which had performed at least two cytokineses (Fig 2C). We then displayed the progression of the DeepCycle pseudotime along the relative cell cycle time, the CC time (Fig 2D). We obtained almost perfect correlations with CC time (Spearman $r = 0.97$, Kendall tau = 0.89; both *P*-values < 0.001), thus validating the reconstructed trajectory of DeepCycle and its ability to capture cell cycle-related features from unsegmented microscopy images in an unsupervised manner.

Reconstructing the relative cell cycle progression opens novel avenues to investigate the cell cycle phenomenon. Our study was enabled by time-lapse imaging for *in situ* analysis of cells tracked over their cell cycle, contrary to previous reports (Gut *et al*, 2015; Eulenberg *et al*, 2017), which used either flow-through microscopy or destructive fixation protocols. The particular challenge that we addressed in this work is that live-cell microscopy provides worse spatial resolution and image quality compared to imaging of fixed cells as in Gut *et al* (2015) who used 40x or confocal microscopy. This puts additional requirements on the computational methods. We demonstrated that using a combination of state-of-the-art deep learning methods for feature extraction and UMAP for manifold learning enabled the reconstruction of a cell cycle trajectory even from 10x microscopy. We also demonstrated the relevance of using the FUCCI2 system to recover a cyclic trajectory that represents continuity even through the cytokinesis moment. Importantly, by using time-lapse microscopy

and automated cell tracking, we have been able for the first time to validate the progression of the recovered trajectory by comparing it to the CC time.

We have developed DeepCycle, a deep learning-based method able to learn a continuous closed cell cycle trajectory aligned with the CC time, from 2.6 million single-cell microscopy images. This achievement illustrates the ability of DeepCycle to extract relevant biological knowledge from cell images and to intuitively represent them in the context of the cell cycle. The cyclic nature of the learned trajectory can provide new insights into the continuity and preserved similarity between the mother and daughter cells. More technically, DeepCycle enables the ability to work with live-cell microscopy unsegmented images as well as reducing requirements on the input images as only the brightfield and Hoechst channels are required for prediction. This frees up fluorescent channels for additional measurements and opens novel ways of identifying cell cycle progression *in situ* in a high repertoire of microscopy experiments. We hope such advances will help to understand not only the cell cycle but also its influence on cell biology in general. We also expect single-cell analysis to benefit from DeepCycle as the cell cycle is a known confounder. DeepCycle would not only help predict the cell cycle progression of each cell but would also alleviate the need for additional cell manipulation such as population synchronization. Ultimately, we expect DeepCycle to become a valuable tool in microscopy and spatial single-cell analyses.

# Materials and Methods

### MDCKII cell culture

A fresh culture of MDCKII cells was grown for 48 h in the growth medium DMEM + 10% FBS at 37°C, 5% $CO_2$ in a 100 × 21 mm Petri dish (Thermo Fisher Scientific) until a confluence of 90%. The growth medium was replaced with 37°C phosphate-buffered saline (PBS) 1×. The cells were then treated with 10 ml trypsin-EDTA 10% (Sigma Aldrich) for 5 min at 37°C, 5% $CO_2$. The cells were resuspended in a fresh 37°C growth medium and split at a 1:10 (v:v) ratio in a new Petri dish and left to grow at 37°C 5% $CO_2$ until a confluence of 30%. The cells were washed with 37°C PBS 1× three times and incubated with Hoechst 33342 (1 μg/ml in PBS 1×) for 45 min at 37°C, 5% $CO_2$. The cells were washed again three times with 37°C PBS 1× and finally put back in a fresh 37°C growth medium. The cells were then left at 37°C, 5% $CO_2$ for 1 h prior to time-lapse imaging.

### Time-lapse microscopy

Cells were observed with the Nikon Ti-E inverted microscope (Nikon Instruments) using the CFI Plan APO 10× Lambda objective (NA 0.45) (Nikon Instruments) and the sCMOS PCO edge 4.2 CL camera (pixel size = 0.66 μm) in brightfield and fluorescence (Hoechst: 395 nm, mAG: 470 nm, and mKO2: 555 nm). The microscope was controlled using the Nikon NIS-Elements software. A custom-built incubation chamber mounted on the microscope was set to 37°C 60% humidity and 5% $CO_2$ for at least 1 h before acquisition. The Petri dish containing MDCKII cells in their growth medium at 37°C was mounted on the microscope stage. We

designed a custom time-lapse acquisition pipeline in the Nikon NIS-Elements software using the JOBS function. For 33 h, every 10 min, the microscope acquires a tiled acquisition of 64 fields of view, organized in a grid pattern (8 × 8 fields) with 10% surface overlap between them. The whole imaged area corresponds approximately to 8,000 × 8,000px or 5 × 5 mm. For each tile position, a software-based autofocus is performed and images are acquired in the brightfield (exposure time, et = 7 ms), Hoechst (et = 7 ms), GFP (et = 40 ms), and Cy3 (et = 40 ms) channels. Stitching of tiled frames was performed using the Fiji stitching plugin (Preibisch et al, 2009).

## Cell tracking and cell filtering

We used the TrackMate Fiji plugin (Tinevez et al, 2017) for cell tracking using their Hoechst fluorescence signal. We obtained approximately 52,500 tracks and 2.6 million individual cell detections across all time-lapse frames. We manually filtered the tracks by inspecting the microscopy files and obtained about 1,000 and 50 tracks with one and two cytokinesis events, respectively. The filtering was performed as follows: We selected tracks for which a division event was recorded, which we identified by a new pair of coordinates appearing over time. As the appearance of a new pair of coordinates in a track can be erroneous (for example, another cell coming too close to the tracked cell can be identified as a daughter cell once it distances itself afterward), manual curation of these tracks was required. For each track, an image from the Hoechst channel enclosing the whole track was generated for every time point, allowing manual examination of the cell over time and its potential divisions. Our GitHub repository (see "Code and Data Availability") contains the code that enables manual inspection of the tracks with potential divisions as well as usage instructions.

## DeepCycle design and training

The model is a custom design of a deep convolutional neural network classifier. The structure of the DeepCycle neural net is shown in Appendix Fig S1. It operates as follows: A 2-channel cell image is transformed into a 3-channel image using a 1 × 1 convolutional layer and fed to the ResNet-34 network pre-trained on ImageNet. Intermediate Conv4 activations, after average pooling, are fed into a fully connected layer and a softmax layer. The Conv5 block from ResNet-34 is not used. The final softmax layer generates the probabilities of a cell to belong to each of the virtual classes. The output of the fully connected layer represents a 4-component cell feature vector used in UMAP visualization.

To obtain the virtual classes, we split the two-dimensional FUCCI2 plane into four pseudo-quadrants representing four virtual classes. It is important to note that these classes do not approximate cell cycle phases but rather provide discrete labels derived from the FUCCI2 intensities whose predictions are used as a cost function during the network training. The plane is split so that the four classes are balanced, that is, contain approximately the same number of cells (Fig 1C, classes 1, 2, 3, and 4 contain 660,827, 679,534, 676,216, and 667,876 cells, respectively). We have experimented with larger numbers of virtual classes but 4 proved to provide the best feature vectors in terms of cell cycle representation. The classification model is trained with the categorical cross-entropy loss using stochastic gradient descent. In this work, we use the inputs to the softmax layer (logits) as deep features, so a cell is assigned with a four-dimensional descriptor. We trained the deep convolutional neural network on 1,083 curated single division tracks and validated the results on 50 double division tracks across 200 microscopy frames. Therefore, the training set contains approximately 200,000 cell images, and the evaluation set contains approx. 10,000 cell images. Each cell is represented by a 2-channel patch (Brightfield and Hoechst), 31.68 μm × 31.68 μm (48 pixels × 48 pixels). All channels are standardized channel-wise with the frame mean and standard deviation. The model is trained with learning rate $10^{-3}$ and batch size 32. We achieve convergence by the 10th epoch of training. One full training epoch on 200,000 cell images (1,000 tracks images for 200 tracks) took about 15–20 min on a GTX-980.

## DeepCycle framework design

DeepCycle pipeline implements the following steps.

1   Individual cell encoding with a four-dimensional representation. We project rectangular 2-channel images of cells (BF/Hoechst) into four-dimensional vectors using virtual classes on the FUCCI2 plane as targets (see Fig 1B for the DeepCycle model overview; Appendix Fig S1 for a more detailed description of the DeepCycle model; Fig 1C for virtual classes; Appendix Fig S2 for virtual class classification accuracies and confusion matrices).

2   UMAP embedding of the encoded data (preprint: McInnes et al, 2018). On this step, we decrease the representation dimensionality, obtaining a two-dimensional cloud of points of individual cells. The cloud features a visible circular structure resembling the cell cycle (Fig 1D)

3   Self-organizing maps (SOM) were used to find a 1-dimensional closed path in the 2-dimensional point cloud and to cluster the data points along the path. We apply a modified version of SOMPY, an open source Python Library for Self Organizing Map (SOM), where we implement a cylindrical map to create a trajectory with circular structure and spherical initialization of the neurons. One important SOM feature is its ability to preserve topological properties of the data. In earlier experiments, we used a simple circular approximation of cell cycle trajectory that required manual selection of the point cloud center and did not account for data density and structure. We found trajectory approximation using SOM superior, being a fully automatic and data-bound approach. The resulting trajectory is referred to as DeepCycle trajectory and its progression as DeepCycle pseudotime (Fig 2A, Appendix Fig S4).

4   The Spearman correlation and Kendall's tau were employed as validation metrics for the model to estimate the progression of the CC time (Fig 2D). Both methods measure the correspondence of the two variables and provide an adequate statistical estimation. The SciPy v0.15.1 Python package implementations of both measures have been used in this study.

Pilot experiments have been performed to evaluate the performance of DeepCycle when trained on 1-channel input images with either the brightfield (Appendix Fig S5) or the Hoechst (Appendix Fig S6) channel. We also explored whether computing fluorescence intensities from segmented nuclei would improve the CC time prediction (Appendix Fig S7). Finally, to demonstrate the relevance of using four

virtual classes in an unsupervised fashion, we compare the present results to a DeepCycle model trained on manually labeled cell cycle phases from the FUCCI2 intensities (Appendix Fig S8). Discussion of these pilot experiments is present in the figure captions.

**UMAP manifold and pseudotime estimation**

To generate the manifold projection of 2.6 million cell images, we used a Python implementation of the Uniform Manifold Approximation and Projection (UMAP) algorithm (https://github.com/lmcinnes/umap, v0.3.10) on the feature vector generated from the last fully connected layer of DeepCycle ($n$_components = 2, $n$_neighbors = 300, min_dist = 0.05, and metric="correlation"). To estimate the DeepCycle pseudotime, we used self-organizing maps (SOM) to find the 1-dimensional closed path inside an annular cloud of points on the 2-dimensional UMAP plane. We use a modified version of SOMPY, an open source Python Library for SOM, where we introduce a cylindrical map shape to create a map with circular structure and implement spherical initialization of the neurons.

## Code and data availability

The data are deposited at EBI BioStudies https://www.ebi.ac.uk/biostudies/studies/S-BSST323. The trained deep learning models as well as the code for obtaining the DeepCycle representation and for retraining models from new images are available at https://github.com/alexandrovteam/DeepCycle.

**Expanded View** for this article is available online.

## Acknowledgements
This work was supported by the European Research Council (grant No 773089), the EU project ATTRACT (grant No 777222), and Russian Foundation for Basic Research (RFBR) grant 18-54-74005 EMBL_t. We thank Leticia Rodríguez Montes (EMBL) for helping with cell culture and live-cell imaging protocols and Lars Hufnagel (EMBL) for providing the MDCKII FUCCI cell line. We also thank Rainer Pepperkok, Beate Neumann, and Sabine Reither from the Advanced Light Microscopy Facility of EMBL for providing the imaging solutions used in this study as well as their technical support with the live-cell imaging. Open access funding enabled and organized by Projekt DEAL.

## Author contributions
LR and ARi acquired and analyzed the live-cell imaging data (cell tracking and manual curation). ARa designed and trained the DeepCycle model. LR and ARa performed data analysis. LR, TA, ARa, and SN wrote the manuscript. SN and TA supervised and coordinated the work.

## Conflict of interest
The funders had no role in the study design, data collection and interpretation, or the decision to submit the work for publication. The authors declare that they have no conflict of interest.

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
