## [Review Process File · Molecular Systems Biology]

DeepCycle reconstructs a cyclic cell cycle trajectory from unsegmented cell images using convolutional neural networks

Luca Rappez, Alexander Rakhlin, Angelos Rigopoulos, Sergey Nikolenko, and Theodore Alexandrov
DOI: [10.15252/msb.20209474](https://doi.org/10.15252/msb.20209474)

Corresponding author(s): Theodore Alexandrov (theodore.alexandrov@embl.de)

Review Timeline:

Submission Date:	14th Feb 20
Editorial Decision:	6th Apr 20
Revision Received:	20th Jul 20
Editorial Decision:	18th Aug 20
Revision Received:	2nd Sep 20
Accepted:	8th Sep 20

Editor: Maria Polychronidou

Transaction Report:

Thank you again for submitting your work to Molecular Systems Biology. We have now heard back from the three referees who agreed to evaluate your study. Overall, the reviewers acknowledge that the study addresses a relevant problem and think that the proposed approach could be useful. However, they mention that as it stands the study seems somewhat preliminary and they raise a series of concerns, which we would ask you to address in a major revision.

Without repeating all the points listed below, some of the more fundamental issues raised are the following:

- The reviewers point out that training and validation metrics need to be reported.
- Clear validation needs to be provided to demonstrate that the method works well and can be reliably used by other in future analyses. The advantages and applications of the proposed approach need to be clearly described. Ideally some concrete examples of the insights that can be derived from these analyses should be provided.
- The code and data need to be made available.
- The reviewers raise several technical concerns (e.g. regarding the presented Fucci distribution, a potentially inaccurate representation of the cell cycle in the classification step etc.), which would need to be compellingly addressed.

All other issues raised by the reviewers would need to be convincingly addressed. Please let me know in case you would like to discuss any of the issues raised by the reviewers.

On a more editorial level, we would ask you to address the following issue.

REFEREE REPORTS

Reviewer #1:

In this manuscript the authors train a convolutional neural network to predict the cell cycle phase of MDCK cells using only transmission brightfield and DAPI data.

Overall, I enjoyed reading the manuscript. The concept is interesting, the manuscript is clear, albeit missing some essential information. This is a good use of CNNs to provide information about cell cycle state directly from simple imaging, without the need for complex markers to be present. However, I found that some details of the methodology are vague and/or missing, some of the

validation metrics are absent from the manuscript, and additional insight from the data could be presented with a little extra work.

I have summarized the major and minor points below:

Major

- The DAPI channel seems ill-suited to future applications in live cell imaging. Have the authors tried this approach with other commonly used live cell markers such as H2B-GFP?
- Further, how well does the system work if either the brightfield or DAPI channels are replaced with Gaussian noise? Are there systematic errors of state classification? Another way of thinking about this is, what features are used for specific classifications?
- Are the 2.6 million images used for training all from the same set of approx 1000 cells? Was data augmentation used? One concern is that since many of the images are from sequences of images, they will also cluster in that manner. Have the authors tested their model using a completely unseen dataset?
- I also cannot find any details of the accuracy, precision or recall of the network in the classification task that is used to train the model. In the methods, a validation set is mentioned, but no metrics are reported.
- In the low-dimensional representation, it seems that the G1 phase is largely mixed with the G1/S phase (although this does not prevent the authors from calculating the pseudo-time). Have you tried performing the dimensionality reduction using the previous network layer (512 features) - does this change the output? [As an aside: if you perform UMAP using the Cdt1 and Geminin intensity information only, do these naturally order in a similar manner?]

Minor

- Should cite Bove et al. (2017) MBoC 3215-3228 in the introduction. The authors used timelapse microscopy, convolutional neural networks and single cell tracking to characterise proliferation in MDCK cell populations.
- Are all of the measurements at low cell density? Can you comment on the range of cell densities used?
- Is the feature vector in Fig 1 just the unscaled logits? Or is there a further transformation?
- The authors should provide a link to the source code, and ideally, a trained model

Reviewer #2:

The work by Rappez and colleagues entitled "DeepCycle: Deep learning reconstructs a closed cell cycle trajectory from single-cell images" is a deep-learning based approach to generate a cell-cycle representation that can be used to map microscopy images of single cells into a prediction of their cell-cycle position.

I'm not an expert on the cell-cycle and cannot judge the relevance of the presented work for the research being conducted in that arena. Hence, please read my review with this in mind. One major criticism regarding the manuscript is that I'm not seeing how the presented work or results can benefit others to improve their own research. If such clear benefits exist, should it not be easy for the authors to point this out in a way that any reader sees this potential, independent on their background?

Additional to the manuscript not giving examples of how DeepCycle could be used in productive

ways, it also appears that code and used data is not publicly available. This is deeply concerning for multiple reasons: (i) results cannot be reproduced, (ii) the paper is in many technical points so superficial that I would not be very confident to being able to reproduce an equivalent pipeline (similar maybe yes), (iii) the network will have to be retrained if other data should be used, hence, even if there are important use-cases for DeepCycle that are obvious to others, they can't profit from it unless they invest some serious development effort.

One more thought: in order to train DeepCycle, one need adequate data and has to perform a good quality cell tracking (e.g. with Trackmate, as described in the paper). Only then the training can start. After training, how do I know the training was successful? Is there a way for users to check that? But lets say all worked just fine, again I wonder what the return on invest is. What can researchers now learn from the result? How can DeepCycle now be used in ways that make the invested efforts worth the while?

Here a few of my more detailed comments:

- Abstract misses to mention the training objective. Currently the story is that some training was performed on some amount of data.
- Abstract: "... show its linear progression with the relative cell cycle time ..." - Where is this shown? What does this even mean?
- Mention of a novel 'cell2vec' comes without any citation or explanation of what this means. I happen to know "word2vec" and what the general idea is, but for others this might mean nothing. Additionally, regular word2vec approaches are quite different from the training procedure presented in this manuscript (e.g. typically leading to much higher dimensional encodings than only 4 dimensions).
- Nowhere in the manuscript I found statements about the size of the extracted single cell images.
- Nowhere in the manuscript I found training times.
- Why are the ground truth (GT) labels derived from the GFP and RFP channels called 'virtual classes'? What makes them virtual?
- The excitement about Figure 1D is completely lost on me. Yes, there is something with a big hole in the middle. First of all this is a 2D projection in some non-linear UMAP mapping of the original 4D vector space, but calling this a circle is far fetched. There are many other holes and potentially interesting features that stay unexplored and undiscussed. It also concerns me that the FUCCI2 false coloring is not circular. If the paper would have stated that the color is a red to green gradient from left to right, that would be at least as believable as the claim for it to be circular.
- Figure 2: The difference between B and D is still confusing me. Also... if D shows the average of 50 cells, why would the authors not also show the standard deviation, as they do in B? What do I learn from this figure? As I said, I might really not be the target audience of this paper, but in my opinion the authors fail in making the basics clear.
- I know what a SOM is, but I'm not sure how SOMs are used to find the "DeepCycle trajectory". Why are SOMs used for this task? Is there a reason why a SOM's opinion of this trajectories position is of higher quality than other ways of getting similar ones? Is it a good idea in the first place to focus on any one trajectory? Why are the other positions in the learned encoding not worth being looked at? In general, it is always a good practice to motivate methodological choices.
- At the end of the paper the authors write: "Reconstructing the relative cell cycle progression opens novel avenues to investigate the cell cycle phenomenon." -- the authors should provide at least some examples of such novel avenues.
- Similar in conclusion: "...extract relevant biological knowledge from cell images and to represent them in an intuitive and comprehensible way." I missed to spot any intuitive representation and obviously comprehended little. If anything, the vector encoding raises many questions in my mind

about what it could be used for... or even more fundamentally, what it really represents.

- Supp. Fig. 2: This figure does not require DeepCycle, does it? Why is it there? What is the relevance for the paper? I'm also confused why pre-mitotic and post-mitotic is different. Is this a property of the very specific dataset? Is it (in parts) an artifact induced by all plots being aligned at the moment of cytokinesis?

Supp. Fig. 3: What do we learn from this figure???

- Supp. Fig. 4: what is the violet data shown in the background? What does this Figure try to communicate?

Reviewer #3:

The manuscript presents a deep learning method, called DeepCycle, to predict cell cycle position using brightfield and nuclear labeling in live cell images. The authors use the FUCCI cell model to infer cell cycle position and introduce the Cell2vect transformation method which converts CDT1 and GMNN staining to discrete training labels for deep learning classification. The author used a large set of confocal microscopy, bright field images and time lapse videos. While such a method would indeed address an unmet need in life science research, for example to allow monitoring of the cell cycle in real time during experiments or freeing up more fluorescent channels for other markers, the approach lacks specificity and validation to provide the credibility needed to ensure this could be a tool adopted by the field.

Major comments:

1. DeepCycle performs the classification task of determining cell cycle position at low resolution. They used Cell2vect to convert all available cells into 4 balanced virtual classes (i.e. containing a similar number of cells). First of all, this is not reflective of the biological cell cycle, where cells spend more time in certain phases than others (G1 which is the red cells, usually take up almost half of the total cell cycle time). Meaning that the cells in each of these 4 groups will not always be very similar to each other in terms of cell cycle position. By combating class imbalance in this way, they create an inaccurate representation of the cell cycle, likely causing the model to be less performant in an actual context.

2. No training and validation metrics (loss, accuracy etc) were reported, without such information it is impossible to evaluate the performance of the model. For validation, the authors mentioned that 50 cells which go through cytokinesis at least twice were manually validated. However, there was no specific figure or analysis of how many cells were accurate, at which phase, or analysis on if the model perform equally on cells with longer or shorter division rate etc.

3. Since the authors use two channels (brightfield and dapi) for prediction, it would also be relevant to see numbers of the performance on either one of these channels alone.

4. The cell cycle displayed in Figure 1C Are substantially different from the classical fucci cell fluorescent distribution, as presented for example in the original fucci publication (REF), with clear patterns of 3 clusters for CDT1 dominant (red), transition (yellow), and GMNN dominant (green). The authors should provide an explanation for this.

5. The authors conclude that the advantage of this model is that it can learn continuous trajectories from large populations of unsynchronized cells. The resulting UMAP visualization however, leaves many questions unanswered (1D, 2D). The fucci cells are Red in the beginning of

the cell cycle, turn yellow in the transition to S and are green for most of S phase and G2 phase. In the UMAP the red cells are all at the border of the entire circle, why is this? This does not agree with the conclusion of a closed trajectory? In figure 2D the authors show the trajectory through this UMAP for a selected cell. The authors should analyze the trajectories of all cells in the UMAP, or at a minimum a significant number starting at different red positions in the UMAP. The Fucci Red and Green trajectories measured according to the pseudotime (2B) and real time (2C) also looks quite different, particularly for the red fucci marker. The authors need to present performance metrics for cells in different parts of the cell cycle and analyze the validity of the trajectory, for example, how well do cells match the average track?

Minor comments:

1. The fluorescence intensities of GMNN and CDT1 were calculated as the mean pixel values over a square of 20x20 pixels centered around the centroid of the cell nucleus. This calculation disregards the heterogeneity in nucleus size in the dataset. Perhaps a more refined segmentation of nuclei and the mean of respective regions would provide more precise intensity values and hence more exact downstream analysis.
2. The heterogeneity of FUCCI2 signals observed in pre- and post- mitotic time might be due to the difference cell cycle length mentioned in the main text (20-32h). Would this heterogeneity reduce if all tracked cells are normalized by their division rate?
3. Dapi is normally used for fixed cells and hoescht for live cell staining of the nucleus, due to its membrane permeant properties. The authors write dapi throughout the main text, but hoescht in the methods. This should be clarified.

We would like to thank all three reviewers for their constructive feedback and their efforts in reading and commenting on our manuscript. We have addressed all the raised comments and believe this significantly improved the manuscript. Our responses are highlighted in blue.

Reviewer #1:

In this manuscript the authors train a convolutional neural network to predict the cell cycle phase of MDCK cells using only transmission brightfield and DAPI data.

Overall, I enjoyed reading the manuscript. The concept is interesting, the manuscript is clear, albeit missing some essential information. This is a good use of CNNs to provide information about cell cycle state directly from simple imaging, without the need for complex markers to be present. However, I found that some details of the methodology are vague and/or missing, some of the validation metrics are absent from the manuscript, and additional insight from the data could be presented with a little extra work.

We are very happy and grateful for the positive feedback! We have edited the manuscript to make the presentation more clear as well as presented additional validation metrics. Below we provide the detailed point by point response.

I have summarized the major and minor points below:

Major

- The DAPI channel seems ill-suited to future applications in live cell imaging. Have the authors tried this approach with other commonly used live cell markers such as H2B-GFP?

Indeed the illumination in the DAPI channel can be toxic if the exposure time is not controlled. We performed the live cell imaging according to the best practice in the field (Spencer *et al*, 2013), which accord to say that the exposure time should be maintained below 525ms for each timepoint, which we did (94ms). Importantly, to clarify a possible confusion, we note that the DAPI naming concerns the channel only. For staining, we employed the live-cell compatible nucleus staining Hoechst. We have replaced it throughout the text to avoid the confusion. To answer the suggestion of the use of H2B-GFP marker, we could not use it in this particular experiment as the GFP channel was dedicated to monitor the mAG-Geminin abundance from the FUCCI construct. This is still a valid comment for other channels and will definitely be explored in future work, thank you.

- Further, how well does the system work if either the brightfield or DAPI channels are replaced with Gaussian noise? Are there systematic errors of state classification? Another way of thinking about this is, what features are used for specific classifications?

Identifying the contribution of individual features to the classification accuracy is indeed a topic of high interest. Following the reviewer suggestion, we have performed training of the network and classification of cells into four virtual classes using either DAPI or brightfield channel alone.

We found that using just one channel somewhat reduced the classification accuracies (Hoechst alone:0.64, brightfield alone:0.64, both channels:0.73; more details are shown in **Appendix Figure S2**). We note that, using Hoechst alone led to a higher accuracy than using the brightfield suggesting that the Hoechst signal is contributing more than the brightfield channel to cell classification. This is likely due to the known relationship between the DNA content and the cell cycle. Nonetheless, a combination of the Hoechst and brightfield channels is required to obtain the classification results presented in the current manuscript.

- Are the 2.6 million images used for training all from the same set of approx 1000 cells? Was data augmentation used? One concern is that since many of the images are from sequences of images, they will also cluster in that manner. Have the authors tested their model using a completely unseen dataset?

The model was trained on 1000 cells (approx 200,000 cell images) and tested on 2.4 million cell images, which is about 12x larger. Concerning data augmentation, we used it in pilot experiments but found out that it didn't improve the results. This is likely due to the large number of training images (200.000) and their variability.

- I also cannot find any details of the accuracy, precision or recall of the network in the classification task that is used to train the model. In the methods, a validation set is mentioned, but no metrics are reported.

The classification accuracies and confusion matrices are now included in the present manuscript; see **Appendix Figure S2** for more details.

To clarify the validation of the model, the classification of cells into four virtual classes was used for model training but for validation we used the correlation of the DeepCycle pseudo-time (progression through the DeepCycle trajectory) with the cell cycle time (CC time, defined for each cell as its relative time point between two divisions). The latter was inferred for cells which were tracked over their cell cycle with both divisions detected in live-cell imaging. These results are reported in **Figure 2D** and show a nearly perfect correspondence between the CC time and the inferred DeepCycle pseudo time. We have modified the manuscript to present our validation strategy and results in a clearer way.

- In the low-dimensional representation, it seems that the G1 phase is largely mixed with the G1/S phase (although this does not prevent the authors from calculating the pseudo-time). Have you tried performing the dimensionality reduction using the previous network layer (512 features) - does this change the output? [As an aside: if you perform UMAP using the Cdt1 and Geminin intensity information only, do these naturally order in a similar manner?]

We thank Reviewer 1 for this suggestion! We performed the suggested UMAP dimensionality reduction of the output of the 256 layer and the FUCCI fluorescence intensities alone. In both cases, the circular ordering of the cells along their cell cycle was not achieved, but rather obtaining two clusters separating the mAG+/mKO2- and the mAG-/mKO2+ cells (data not

shown). This suggests that an additional fully connected layer is required to transform the data in order to highlight the cell cycle related ordering of the cells in UMAP.

Minor

- Should cite Bove et al. (2017) MBoC 3215-3228 in the introduction. The authors used time lapse microscopy, convolutional neural networks and single cell tracking to characterise proliferation in MDCK cell populations.

Indeed, this is a relevant study and we have added a citation to it in the introduction.

- Are all of the measurements at low cell density? Can you comment on the range of cell densities used?

This is an important point. At the beginning of the time-lapse imaging, the cell confluence was set to be between 20-30%. After 16hrs of proliferation, the cell culture reached approximately 80-90% confluence.

- Is the feature vector in Fig 1 just the unscaled logits? Or is there a further transformation?

The feature vector presented in **Figure 1** is composed of unscaled logits with no further transformation.

- The authors should provide a link to the source code, and ideally, a trained model

We have deposited the source code and the trained model to the GitHub <https://github.com/alexandrovteam/DeepCycle> and added the link in the 'Code availability' section of the methods.

Reviewer #2:

The work by Rappez and colleagues entitled "DeepCycle: Deep learning reconstructs a closed cell cycle trajectory from single-cell images" is a deep-learning based approach to generate a cell-cycle representation that can be used to map microscopy images of single cells into a prediction of their cell-cycle position.

I'm not an expert on the cell-cycle and cannot judge the relevance of the presented work for the research being conducted in that arena. Hence, please read my review with this in mind. One major criticism regarding the manuscript is that I'm not seeing how the presented work or results can benefit others to improve their own research. If such clear benefits exist, should it not be easy for the authors to point this out in a way that any reader sees this potential, independent on their background?

We adapted the introduction and the conclusion to better highlight the added benefits of performing such analysis. Briefly, using DeepCycle:

- enable the combination of label-free cell cycle investigation with pre-existing microscopy-related technologies involving conventional fluorescent dyes.
- will allow the delineation of cell cycle dependent features from other biological reactions, a feature which is of particular importance for single cell technologies as cell cycle acts as a strong confounding factor in most biological processes.
- alleviates the need for cell population synchronization, which is an additional sample manipulation imposing potential cellular stress
- simply offers a new basis for investigating the cell cycle progression from microscopy which is valuable piece of information for fundamental research in general

Additional to the manuscript not giving examples of how DeepCycle could be used in productive ways, it also appears that code and used data is not publicly available. This is deeply concerning for multiple reasons: (i) results cannot be reproduced, (ii) the paper is in many technical points so superficial that I would not be very confident to being able to reproduce an equivalent pipeline (similar maybe yes), (iii) the network will have to be retrained if other data should be used, hence, even if there are important use-cases for DeepCycle that are obvious to others, they can't profit from it unless they invest some serious development effort.

We have deposited the source code and the trained model to the GitHub <https://github.com/alexandrovteam/DeepCycle> and added the link in the 'Code availability' section of the methods.

One more thought: in order to train DeepCycle, one need adequate data and has to perform a good quality cell tracking (e.g. with Trackmate, as described in the paper). Only then the training can start. After training, how do I know the training was successful? Is there a way for users to check that? But lets say all worked just fine, again I wonder what the return on invest is. What can researchers now learn from the result? How can DeepCycle now be used in ways that make the invested efforts worth the while?

We added a new metric, presented in **Figure 2D**, which informs about the success of the model in inferring a meaningful cell cycle trajectory. The metric quantifies the correspondence, using the Spearman correlation and Kendall's tau, of the inferred DeepCycle pseudotime with the real CC time measured from cells undergoing a full cell cycle using live cell imaging.

We answered the added benefits of using the DeepCycle model in the earlier comment of Reviewer 1.

Here a few of my more detailed comments:

- Abstract misses to mention the training objective. Currently the story is that some training was performed on some amount of data.

We judged that the training strategy should be mentioned first in the introduction rather than in the abstract as it is a technical part of the study. The training strategy is presented in the 3rd paragraph of the introduction and illustrated in **Figure 1A**.

- *Abstract: "... show its linear progression with the relative cell cycle time ..." - Where is this shown? What does this even mean?*

The relation between the DeepCycle pseudotime with the relative cell cycle time (CC time) is shown in **Figure 2D** which shows a strong monotone association between them. The DeepCycle pseudotime was calculated as the progression through the DeepCycle trajectory. The CC time was calculated as the relative time between two consecutive cell divisions, obtained for cells which we tracked using live-cell imaging using cell tracking software, verified using manual curation. We however accept the criticism that the term 'linearity' should not be used and have now decided to remove it, replacing with the "correlation" between DeepCycle pseudotime and CC time, that we quantified using the Spearman correlation and Kendall tau (**Figure 2D**). Furthermore, addressing the reviewer comment, we have edited the caption of **Figure 2D** and the manuscript text to improve on the presentation of this result. In particular, in the new text we highlighted that the achieved nearly-perfect correlation between the DeepCycle pseudotime and CC time (Spearman $r=0.97$, $p\text{-value}<0.001$) provides a strong validation of the DeepCycle model.

- *Mention of a novel 'cell2vec' comes without any citation or explanation of what this means. I happen to know "word2vec" and what the general idea is, but for others this might mean nothing. Additionally, regular word2vec approaches are quite different from the training procedure presented in this manuscript (e.g. typically leading to much higher dimensional encodings than only 4 dimensions).*

We agree that the term 'cell2vec' was not properly introduced. We initially planned to use it to refer to our mapping of cells into the low-dimensional space. However, we decided for the DeepCycle name as it better reflects the essence of our approach. We thank the reviewer for spotting this inconsistency and we are sorry we forgot to remove the reference to cell2vec. We have cleaned the manuscript and removed the references to 'cell2vec' everywhere.

- *Nowhere in the manuscript I found statements about the size of the extracted single cell images.*

The cell images, centered around the geometrical center of the cell nuclei, have dimensions of 48x48 pixels, with a pixel size of 0.66 μm thus having dimensions of 31.68x31.68 μm . This information has been added in the 'DeepCycle design and training' section of the methods.

- *Nowhere in the manuscript I found training times.*

We have added this information in the 'DeepCycle design and training' section of the methods. Typically one full training epoch on 200.000 cell images (1000 tracks images for 200 tracks) took about 15-20 min on a GTX-980.

- Why are the ground truth (GT) labels derived from the GFP and RFP channels called 'virtual classes'? What makes them virtual?

The term 'virtual' was chosen to reflect the 'simulated' aspect of these classes. It is important to clarify that such classes do not reflect cell cycle phases and were solely designed for the training of the network, therefore we judged it is relevant to add this adjective to help avoiding such possible confusion. If the term 'virtual' introduces confusion that we have not noticed, we will be happy to change it.

- The excitement about Figure 1D is completely lost on me. Yes, there is something with a big hole in the middle. First of all this is a 2D projection in some non-linear UMAP mapping of the original 4D vector space, but calling this a circle is far fetched. There are many other holes and potentially interesting features that stay unexplored and undiscussed. It also concerns me that the FUCCI2 false coloring is not circular. If the paper would have stated that the color is a red to green gradient from left to right, that would be at least as believable as the claim for it to be circular.

Thank you for raising these concerns about the presentation of our results. We have adapted the text to improve on it in particular in the interpretation of **Figure 1**. Responding to the raised concerns: Firstly, coloring the cells by their FUCCI2 intensities proves that we successfully organised the cells in a relevant way as the red and green fluorescence intensity appear sequentially over time as cells progress through their cell cycle. Secondly, the main projection axis is cyclic. As we understand that the term 'circular' could mislead as it is not a perfect circle but rather a cyclic projection, we modified the text accordingly. The relevance of a cyclic projection is further discussed in the main text. The reason we did not mention the 'smaller holes' is to keep the main story simple. The main trend highlighted by the FUCCI signal is minimally impacted by such structures and therefore motivated us to focus on the 'large hole' which separates the mKO2-Cdt1 prominent from the mAG-Geminin prominent cells.

- Figure 2: The difference between B and D is still confusing me. Also... if D shows the average of 50 cells, why would the authors not also show the standard deviation, as they do in B? What do I learn from this figure? As I said, I might really not be the target audience of this paper, but in my opinion the authors fail in making the basics clear.

Again, thank you for this comment as it helped us improve the presentation of **Figure 2D**.

Responding to the raised concern, we explain our flow of thought: in **Figure 1D** we present the cyclic UMAP projection color coded with the FUCCI intensities. The fact that red and green cells are separated validates the ability of DeepCycle to capture cell cycle related features as such FUCCI intensities sequentially appear along the cell cycle progression. In **Figure 2A**, we

present a trajectory which captures the main cyclic axis of the UMAP projection. We then show the FUCCI fluorescences of all 2.6 million cell images sorted along this trajectory in **Figure 2B**, which illustrates in a more quantitative way the expression of the mKO2-Cdt1 (red) and mAG-Geminin (green) FUCCI 2 cell cycle markers along the DeepCycle trajectory. For validation purposes in **Figure 2D**, we show the progression of the DeepCycle trajectory (the pseudotime) in function of the real cell cycle time (CC time), which is used as ground truth in this context. To obtain the latter, we manually selected 50 cells which have been tracked for a full cell cycle and measured the relative time spent between two cytokinesis (in %) to define their CC time. We then reported the average DeepCycle pseudotime of these 50 cells in function of the CC time as they progressed through their cell cycle. The correlation between the pseudotime and the CC time provides a measure of their correspondence, which we used as a validation metric. Here, we obtained a spearman correlation of 0.97 which we deemed sufficient to validate the reconstructed DeepCycle trajectory.

- I know what a SOM is, but I'm not sure how SOMs are used to find the "DeepCycle trajectory". Why are SOMs used for this task?

Addressing this question, we have edited the 'DeepCycle framework design' section of the Methods. The main reason for using SOM is its ability to preserve topological properties of the data.

Is there a reason why a SOM's opinion of this trajectory's position is of higher quality than other ways of getting similar ones?

In earlier experiments, we used a simple circular approximation of cell cycle trajectory that required manual selection of the point cloud center and did not account for data density and structure. Additionally, we explored other pseudotime inference from 2 dimensional points and SOM gave the best results for finding the expected cyclic trajectory.

Is it a good idea in the first place to focus on any one trajectory? Why are the other positions in the learned encoding not worth being looked at?

In our processing framework, every cell is classified into a SOM cluster, which are sequentially positioned along the trajectory shown in **Figure 2A**. This means that there are no positions in the UMAP projection that are not incorporated in the downstream analysis.

In general, it is always a good practice to motivate methodological choices.

We agree and hope we have properly addressed this comment in the revised method section.

- At the end of the paper the authors write: "Reconstructing the relative cell cycle progression opens novel avenues to investigate the cell cycle phenomenon." -- the authors should provide at least some examples of such novel avenues.

We have edited the text to put more emphasis on what the method enables in terms of novel combinatorial approaches with specific examples added to the Introduction and Conclusion.

- *Similar in conclusion: "...extract relevant biological knowledge from cell images and to represent them in an intuitive and comprehensible way." I missed to spot any intuitive representation and obviously comprehended little. If anything, the vector encoding raises many questions in my mind about what it could be used for... or even more fundamentally, what it really represents.*

Responding to your comment, we assumed that for a biologist it can be more intuitive to see in different points of the cell cycle to be organized in a circular trajectory, as this is an oscillating process. Moreover, having all cells visualized within one plane (e.g. UMAP) is popular and well-accepted in modern single-cell analyses.

Nevertheless, we heard your concern and adapted the text by using "intuitively represent them in the context of the cell cycle" instead.

- *Supp. Fig. 2: This figure does not require DeepCycle, does it? Why is it there? What is the relevance for the paper?*

Appendix Figure S3 (previously Supp Fig 2) shows the intermediate data used for creating a curated set of tracked cells which were used for calculations of the CC time and warping the DeepCycle time. Specifically, **Appendix Figure S3** shows the heterogeneity of cells in their commitment to the second division. Seeing such a strong heterogeneity motivated us to collate a curated set of 50 cells for which two subsequent divisions were detected and manually confirmed.

I'm also confused why pre-mitotic and post-mitotic is different. Is this a property of the very specific dataset? Is it (in parts) an artifact induced by all plots being aligned at the moment of cytokinesis?

It is a natural property of cells where some engage in their cell cycle to generate new daughter cells while others enter a quiescent state, in which the cell cycle re-entry is delayed. Most cellular systems are composed of both states and it is not possible to distinguish them using the FUCCI2 system. For this reason, when aligning the tracks at the moment of cytokinesis, the pre-mitotic time is only composed of cells engaged in their cell cycle while the post-mitotic time is a combination of dividing and quiescent cells, thus composing a deformed FUCCI trend over time. Additionally, each cell spends a different amount of time dividing, making the fluorescence trends confounded when projected along time. For this reason, we defined the CC time as the relative time spent between two cytokinesis when validating the DeepCycle trajectory as presented in **Figure 2D**.

- *Supp. Fig. 3: What do we learn from this figure???*

This figure explains in better details the construction of the DeepCycle trajectory. The captions help the reader understand which algorithm was used and what it really represents. Indeed, the message of this figure can be interpreted as redundant to **Figure 2A-B**, but also provides more detailed technical insights. The captions have been updated to improve clarity.

- *Supp. Fig. 4: what is the violet data shown in the background? What does this Figure try to communicate?*

The **Appendix Figure S4A** highlights the progression of the DeepCycle trajectory on the FUCCI fluorescence intensity plane (Cdt1-RFP in y axis and Geminin-GFP in x axis). In addition to the DeepCycle trajectory the average progression of the 50 manually curated cells is presented on this plane. The purple background represents the cell count for the 2D histogram performed on that plane, the colorbar with the respective values was missing and has been added. In the **Appendix Figure S4B**, represents the teaching points used of the 2D non-linear warping of the trajectory to the average track, in order to express the trajectory progression in terms of CC time. The captions have been updated to improve clarity.

Reviewer #3:

The manuscript presents a deep learning method, called DeepCycle, to predict cell cycle position using brightfield and nuclear labeling in live cell images. The authors use the FUCCI cell model to infer cell cycle position and introduce the Cell2vect transformation method which converts CDT1 and GMNN staining to discrete training labels for deep learning classification. The author used a large set of confocal microscopy, bright field images and time lapse videos. While such a method would indeed address an unmet need in life science research, for example to allow monitoring of the cell cycle in real time during experiments or freeing up more fluorescent channels for other markers, the approach lacks specificity and validation to provide the credibility needed to ensure this could be a tool adopted by the field.

Major comments:

1. DeepCycle performs the classification task of determining cell cycle position at low resolution. They used Cell2vect to convert all available cells into 4 balanced virtual classes (i.e. containing a similar number of cells). First of all, this is not reflective of the biological cell cycle, where cells spend more time in certain phases than others (G1 which is the red cells, usually take up almost half of the total cell cycle time). Meaning that the cells in each of these 4 groups will not always be very similar to each other in terms of cell cycle position. By combating class imbalance in this way, they create an inaccurate representation of the cell cycle, likely causing the model to be less performant in an actual context.

This particular point was raised by other reviewers and we hope we answered this concern with our new analysis presented in **Appendix Figure S11**. In short, we have considered using 3 virtual classes approximating the cell cycle phases which can be identified with the FUCCI2

construct: G0/G1-S-G2/M. However, the obtained trajectory showed lower correlation between the pseudotime and CC time.

In detail, instead of choosing four balanced virtual classes, we separated the FUCCI intensity plane in three classes approximating the cell cycle phases (**Appendix Figure S11A, B** for training accuracies). Such thresholds were set manually to match the cell cycle phases as presented in Supp. Fig S1 of (Streichan *et al*, 2014); see <https://www.pnas.org/content/pnas/suppl/2014/03/27/1323016111.DCSupplemental/pnas.201323016SI.pdf>. The DeepCycle network has been modified to output a 3 features vector visualized in 2D with UMAP (**Appendix Figure S11C**). The UMAP projection still produces a cyclic projection which separates the mAG+ from the mKO2+ cells (**Appendix Figure S11D**) highlighting the successful identification and interpretation of cell cycle related features from the network. We constructed a new Deep Cycle trajectory on this UMAP which after time calibration, presented a high correlation (spearman=0.93, Kendall's tau=0.79) with the real CC time measured from the cells tracked along their entire cell cycle (**Appendix Figure S11E**). Since the correlation values are smaller than the ones obtained using the four balanced classes (Spearman $r = 0.97$, Kendall tau=0.9), we decided to use the four balanced classes. Moreover, using four virtual classes does not require supervised partitioning of the fluorescence plane.

2. No training and validation metrics (loss, accuracy etc) were reported, without such information it is impossible to evaluate the performance of the model. For validation, the authors mentioned that 50 cells which go through cytokinesis at least twice were manually validated. However, there was no specific figure or analysis of how many cells were accurate, at which phase, or analysis on if the model performed equally on cells with longer or shorter division rate etc.

We thank Reviewer 3 for raising this concern. In the revised manuscript, we report the confusion matrix for the virtual class prediction; see **Appendix Figure S2**. We would like to emphasize that, although predicting virtual classes was used for training the network, the aim is not achieving the best prediction but rather producing a low-dimensional representation of the cells enabling the reconstruction of a cell cycle trajectory. In the revised manuscript, we have introduced an objective measure of quality of this reconstruction, namely the correlation between the DeepCycle pseudotime (progression along the DeepCycle trajectory) and the CC time (relative time between two subsequent divisions). We report the correlation values in **Figure 2D**.

3. Since the authors use two channels (brightfield and dapi) for prediction, it would also be relevant to see numbers of the performance on either one of these channels alone.

Indeed, such suggestions have been raised by other reviewers and the class prediction accuracies are reported in **Appendix Figure S2**. For the results of the full analysis (including also the trajectory and the correlation between the DeepCycle time and CC time), see **Appendix Figure S8** for the case of using brightfield channel alone, and **Appendix Figure S9** for the case of using the Hoechst channel alone.

4. The cell cycle displayed in Figure 1C Are substantially different from the classical fucci cell fluorescent distribution, as presented for example in the original fucci publication (REF), with clear patterns of 3 clusters for CDT1 dominant (red), transition (yellow), and GMNN dominant (green). The authors should provide an explanation for this.

Indeed their distribution does not match what we observed. We assume Reviewer 3 refers to (Sakaue-Sawano *et al*, 2008), **Figure 1G**. We have two hypotheses as to why our distributions are different from this paper: firstly the cell type is different, we used MDCKII with the FUCCI construct, first presented by (Streichan *et al*, 2014). Secondly, the method of fluorescence quantification is different as we used a wide field microscope but the quantification presented by (Sakaue-Sawano *et al*, 2008) was done using a fluorescence-activated cell sorter. Both of these reasons could impact the distribution of the cell fluorescence intensities. On the other hand, we prove that such differences from the classical FUCCI intensities presented in the original paper do not prevent the ability of DeepCycle to resolve the three main FUCCI clusters (mKO2+/mAG- (red); mKO2+/mAG+ (yellow); mKO2-/mAG+(green)) as well as their ordering, as shown in **Figure 2A** and **B**.

5. The authors conclude that the advantage of this model is that it can learn continuous trajectories from large populations of unsynchronized cells. The resulting UMAP visualization however, leaves many questions unanswered (1D, 2D). The fucci cells are Red in the beginning of the cell cycle, turn yellow in the transition to S and are green for most of S phase and G2 phase. In the UMAP the red cells are all at the border of the entire circle, why is this? This does not agree with the conclusion of a closed trajectory?

Thanks to the comment of Reviewer 3 we realized we had a technical issue with the color-coding of the cells as they were presented in **Figure 1D** and **Figure 2A**. The cells are now rightly color-coded in the present manuscript and reflect the expected sequence of FUCCI markers during the cell cycle: mKO2-/mAG- (dark, colorless) for cells which just divided, mKO2+/mAG- (red) for cells in G₀/G₁ phases, mKO2+/mAG+ (yellow) for cells at their entry in S phase, and mKO2-/mAG+ for cells in S/G₂/M phases.

In figure 2D the authors show the trajectory through this UMAP for a selected cell. The authors should analyze the trajectories of all cells in the UMAP, or at a minimum a significant number starting at different red positions in the UMAP.

We apologize for the possible confusion associated with this visualization. Contrary to the understanding of Reviewer 3, it is not the trajectory of an individual cell but rather a common trend shared by all cells across the UMAP projection which defines a main cyclic trajectory. The **Figure 2B** shows the mean (+/- standard deviation) of the FUCCI fluorescence intensities of all cells projected along this trajectory which recovers the sequential appearance of the FUCCI markers along its progression.

The Fucci Red and Green trajectories measured according to the pseudotime (2B) and real time (2C) also look quite different, particularly for the red fucci marker. The authors need to present performance metrics for cells in different parts of the cell cycle and analyze the validity of the trajectory, for example, how well do cells match the average track?

The main reason for the displayed differences in Figure 2B and 2C is that the **Figure 2C** shows the Fucci trends for a curated set of cells which are normalized to the CC time (relative progression of real time between two cytokineses, measured by live-cell imaging). On the other hand the **Figure 2B** shows the Fucci trend along the reconstructed trajectory presented in **Figure 2A**. The differences in the DeepCycle pseudotime and the CC time lead to differences in the Fucci intensities projected along either one or the other. We have now updated **Figure 2D** by removing the Fucci trends to avoid this confusion.

To show how well individual cells match the average track in different parts of the cell cycle (as requested by the Reviewer 3), we have added to **Figure 2D** the confidence intervals (+/- standard deviation) around the mean DeepCycle pseudotime.

Minor comments:

1. The fluorescence intensities of GMNN and CDT1 were calculated as the mean pixel values over a square of 20x20 pixels centered around the centroid of the cell nucleus. This calculation disregards the heterogeneity in nucleus size in the dataset. Perhaps a more refined segmentation of nuclei and the mean of respective regions would provide more precise intensity values and hence more exact downstream analysis.

We thank Reviewer 3 for such suggestions. Following this suggestion, we have performed an additional analysis based on cell nuclei segmentation; see the results in **Appendix Figure S10**. We used the machine learning based method StarDist (Schmidt *et al*, 2018) to obtain the segmentation mask of all tracked nuclei, which enabled the computation of the mean Fucci fluorescence intensities over the entire nucleus area. Training the network on nuclei segmentation based fluorescence intensities does slightly improve the results (Spearman $r=0.98$) compared to computing such fluorescence intensities from a 20x20 pixel square area centered on the cell nuclei geometric centers (Spearman $r=0.97$). However, in light of such minor improvement, for simplicity and ease of use, as well as to avoid the potential propagation of errors in nuclei segmentation, we decided to keep the current approach which does not rely on segmentation.

2. The heterogeneity of Fucci2 signals observed in pre- and post- mitotic time might be due to the difference cell cycle length mentioned in the main text (20-32h).

Indeed, the reasons for the post-mitotic time being more heterogeneous (more variance in both fluorescence channels) are two-fold: 1- as the Reviewer 3 mentioned, the varying growth rate between cells, which also occurs for the same cell between different cell cycles and 2- that a certain proportion of cells are not re-engaging in the cell cycle after their cell division. Thus

producing a heterogeneous population of cells in the post-mitotic times composed of quiescent and dividing cells. Whereas the pre-mitotic time is naturally composed of cells committed to divide, as they all have been selected for that particular event.

Would this heterogeneity reduce if all tracked cells are normalized by their division rate?

This heterogeneity indeed decreases if the fluorescent trends are reported in function of the CC time (relative time progression between two cytokineses). The main limitation is that in order to obtain the CC time, two divisions have to be monitored, which was the case for a very small minority of the cells tracked (only 50 cells were manually validated as having two cell divisions).

3. Dapi is normally used for fixed cells and hoechst for live cell staining of the nucleus, due to its membrane permeant properties. The authors write dapi throughout the main text, but hoechst in the methods. This should be clarified.

Indeed, Hoechst was used and not DAPI. We referred to DAPI as a channel name as it is often referred to in the literature. We understand the confusion and replace DAPI by Hoechst throughout the whole manuscript.

References relevant for the responses:

Sakaue-Sawano A, Kurokawa H, Morimura T, Hanyu A, Hama H, Osawa H, Kashiwagi S, Fukami K, Miyata T, Miyoshi H, Imamura T, Ogawa M, Masai H & Miyawaki A (2008) Visualizing spatiotemporal dynamics of multicellular cell-cycle progression. *Cell* **132**: 487–498

Schmidt U, Weigert M, Broaddus C & Myers G (2018) Cell Detection with Star-convex Polygons. *arXiv [cs.CV]* Available at: <http://arxiv.org/abs/1806.03535>

Spencer SL, Cappell SD, Tsai F-C, Overton KW, Wang CL & Meyer T (2013) The proliferation-quiescence decision is controlled by a bifurcation in CDK2 activity at mitotic exit. *Cell* **155**: 369–383

Streichan SJ, Hoerner CR, Schneidt T, Holzer D & Hufnagel L (2014) Spatial constraints control cell proliferation in tissues. *Proc. Natl. Acad. Sci. U. S. A.* **111**: 5586–5591

Thank you for sending us your revised manuscript. We have now heard back from the three reviewers who were asked to evaluate the revised study. As you will see below, the reviewers think that the study has largely improved after the performed revisions. However, reviewer #3 lists a couple of remaining issues, which we would ask you to address in an exceptional second round of revision.

Moreover, we would ask you to address the following editorial issues.

Reviewer #1:

I am satisfied that the changes made to the manuscript adequately address my concerns. As such, I find the manuscript suitable for publication.

Reviewer #2:

I am very positively surprised how the relatively few edits in the paper could actually make it quite a bit better. Naturally I am still not an expert in the cell-cycle field and parts of the manuscript are still over my head, but after reading the rebuttal to my review and the other 2 reviews gave me additional insights that allow me to value the presented method and results more.

I find it slightly concerning that each reviewer pointed at some fundamental flaws of the paper that caused the authors to change/add names, colors, figure elements, explanations, and other data that was previously missing. I would have expected that the authors see some of these very basic flaws themselves by simply reading the manuscript another time from top to bottom before submitting.

Nevertheless, I did not find any obvious remaining problems in the manuscript, and felt that the overall readability was improved.

Reviewer #3:

This is a revised version of a report describing DeepCycle, a deep learning method that classified live cell images into 4 virtual classes and used the trained feature map to recreate a cell cycle trajectory. Compared to the original submission, the authors have changed the architecture of DeepCycle, corrected UMAP trajectory, included additional 1-channel training, and added a performance measurement on validation set. Overall, this was a clearer explanation of the method and a significant improvement of the manuscript. However, we have two major comments that should be addressed before we can recommend this manuscript for publication in molecular systems biology.

Major comment:

The remaining major comment pertains to the correlation between DeepCycle pseudotime and observed cell cycle time. Figure S5 describes that the pseudotime was warped onto real cell cycle time of 50 cells, as "The trajectory constructed from the UMAP does not progress linearly with the real time". If these are the same manually annotated 50 cells used for validation, then the correlation reported in figure 2D (Spearman $r = 0.97$ and Kendall tau = 0.9) cannot be viewed as unbiased validation score. We suggest that if the warping is necessary, the authors should divide the 50 cells into 2 subsets, one to fit the UMAP to cell cycle time, and another for validation and only report the later as such. Or the authors can use a different set of cells (that are NOT in the validation set) for warping.

The impact of this study would come from a tool that other researchers can apply to their own studies. At current there is a lot of manual validation (tracking cells over the cell cycle) performed to ensure that the model is working as desired. The authors should describe how to use this model, and what minimal validation is recommended before using this tool on other image datasets.

Minor comments:

1. In the DeepCycle design and training section, it was stated that the model was trained on 2-channel 48x48px patches of filtered single cells resulting from the TrackMate FIJI plugin. Questions still remained as to how much extra work needed if other researchers would potentially employ DeepCycle for their data.
2. Please specify if the confusion matrix and other performance scores are calculated on test data, or training data.
3. In figure 2B, how can the normalized values be outside of [0,1] range?
4. In figure S9A&B, why would the 'clean' virtual classe 1, which consists of cells high in CDT1 and therefore around G1, perform the worst, while more 'mixed' classes performed better?

Reviewer #1:

I am satisfied that the changes made to the manuscript adequately address my concerns. As such, I find the manuscript suitable for publication.

We are very pleased that Reviewer 1 is satisfied by the revised manuscript.

Reviewer #2:

I am very positively surprised how the relatively few edits in the paper could actually make it quite a bit better. Naturally I am still not an expert in the cell-cycle field and parts of the manuscript are still over my head, but after reading the rebuttal to my review and the other 2 reviews gave me additional insights that allow me to value the presented method and results more.

I find it slightly concerning that each reviewer pointed at some fundamental flaws of the paper that caused the authors to change/add names, colors, figure elements, explanations, and other data that was previously missing. I would have expected that the authors see some of these very basic flaws themselves by simply reading the manuscript another time from top to bottom before submitting.

Nevertheless, I did not find any obvious remaining problems in the manuscript, and felt that the overall readability was improved.

We thank Reviewer 2 for their positive and constructive feedback which really helped improve the manuscript.

Reviewer #3:

This is a revised version of a report describing DeepCycle, a deep learning method that classified live cell images into 4 virtual classes and used the trained feature map to recreate a cell cycle trajectory. Compared to the original submission, the authors have changed the architecture of DeepCycle, corrected UMAP trajectory, included additional 1-channel training, and added a performance measurement on validation set. Overall, this was a clearer explanation of the method and a significant improvement of the manuscript. However, we have two major comments that should be addressed before we can recommend this manuscript for publication in molecular systems biology.

We thank Reviewer 3 for providing such an extensive review and suggesting improvements.

Major comment:

The remaining major comment pertains to the correlation between DeepCycle pseudotime and observed cell cycle time. Figure S5 describes that the pseudotime was warped onto real cell

cycle time of 50 cells, as "The trajectory constructed from the UMAP does not progress linearly with the real time". If these are the same manually annotated 50 cells used for validation, then the correlation reported in figure 2D (Spearman $r = 0.97$ and Kendall tau = 0.9) cannot be viewed as unbiased validation score. We suggest that if the warping is necessary, the authors should divide the 50 cells into 2 subsets, one to fit the UMAP to cell cycle time, and another for validation and only report the later as such. Or the authors can use a different set of cells (that are NOT in the validation set) for warping.

We thank Reviewer 3 for raising this particular point. We evaluated whether warping the DeepCycle pseudotime onto the CC time was necessary. We found that without time warping we obtain a Spearman r of 0.972 and a Kendall tau of 0.89, versus 0.973 and 0.9 respectively for a time calibrated DeepCycle pseudotime. Given such little improvement on the correlation values and for general clarity of the manuscript, we decided to not warp the trajectory onto real time as it requires additional efforts from the user with only a minor improvement of the CC time prediction. We updated Figure2D showing progression of the DeepCycle pseudotime, computed from the UMAP projection without any time warping, as a function of the CC time, as measured from live cell imaging for 50 manually curated cells. Additionally we removed the Appendix Figures S5-7 which illustrated the strategy for time warping and its impact on the measured FUCCI trends. We also adapted the 'DeepCycle framework design' section of the Methods to these modifications.

The impact of this study would come from a tool that other researchers can apply to their own studies. At current there is a lot of manual validation (tracking cells over the cell cycle) performed to ensure that the model is working as desired. The authors should describe how to use this model, and what minimal validation is recommended before using this tool on other image datasets.

We thank Reviewer 3 for this suggestion. We updated the 'Cell tracking and cell filtering' section of the Methods to better reflect the manual work performed prior to the network training. We also updated our GitHub repository (<https://github.com/alexandrovteam/DeepCycle>) by providing a script for manual curation of the TrackMate results, adding input arguments, and expanding the instructions accordingly (see commits on GitHub: link1, link2, link3).

Minor comments:

1. In the DeepCycle design and training section, it was stated that the model was trained on 2-channel 48x48px patches of filtered single cells resulting from the TrackMate FIJI plugin. Questions still remained as to how much extra work needed if other researchers would potentially employ DeepCycle for their data.

We answered this question in the updated instructions provided at the GitHub repository (<https://github.com/alexandrovteam/DeepCycle>) and provided a Python script TrackMate_filter.py that enables manual curation of the tracks.

Typically, the TrackMate FIJI plugin provides a table containing nuclei locations for all detected cells and their assignment with a track. We automatically filtered these results by selecting tracks for which a division event has been recorded, which we automatically detected as a new pair of coordinates appearing over time. As the appearance of a new pair of coordinates in a track can be erroneous (for example, another cell coming too close to the tracked cell can be identified as a daughter cell once it distances itself afterwards), manual curation of these tracks is required. For each track, an image from the Hoechst channel enclosing the whole track is generated for every time point, allowing manual examination of the cell over time and its potential divisions. Both the track filtering and image generation for manual curation are performed by a custom python script called 'TrackMate_filter.py' which has been added to the [GitHub repository](https://github.com/alexandrovteam/DeepCycle/blob/master/src/TrackMate_filter.py) (https://github.com/alexandrovteam/DeepCycle/blob/master/src/TrackMate_filter.py). The track IDs where either one or two division events have been identified are taken as input by DeepCycle to initiate the training as well as the validation of the model.

2. Please specify if the confusion matrix and other performance scores are calculated on test data, or training data.

The confusion matrices as well as other performance scores were calculated on test data. This information has been added to the figure legends.

3. In figure 2B, how can the normalized values be outside of [0,1] range?

Thank you for raising our attention to this. It came from an error in the code, where the values were rescaled so that the mean values were ranging from 0 to 1. We now corrected this error and reported the mean +/- std of values which were all scaled between 0 and 1.

4. In figure S9A&B, why would the 'clean' virtual classe 1, which consists of cells high in CDT1 and therefore around G1, perform the worst, while more 'mixed' classes performed better?

We thank Reviewer 3 for raising this very interesting observation. We assume the Reviewer 3 refers to virtual class 2 from Appendix Figure S6B (figure numbers have been reordered). Indeed the classification accuracy is worst for this class when DeepCycle was trained on single Hoechst channel images (0.15, as shown on Appendix Figure S6C or S2C) but also on the model trained on single Brightfield channel images (0.0023, Appendix Figure S2B) or on dual Brightfield + Hoechst channels images (0.17, Appendix Figure S2A). We are not sure why this is the case and could not find a plausible explanation for this particular observation. We would like to add that virtual class prediction accuracy was not the objective of the network but rather a metric used during training, and the goal of the entire process was to provide a feature vector that would meaningfully organize the cells in the UMAP space.

Thank you again for sending us your revised manuscript. We are now satisfied with the modifications made and I am pleased to inform you that your paper has been accepted for publication.

Corresponding Author Name: Theodore Alexandrov

Manuscript Number: MSB-20-9474